# Impact of the COVID-19 pandemic on human-nature relations in a remote nature-based tourism destination

Evert Mul ¤*, Francisco Javier Ancin Murguzur, Vera Helene Hausner

Institute of Arctic and Marine Biology, UiT–The Arctic University of Norway, Tromsø, Norway

¤ Current address: Norwegian Institute for Nature Research (NINA), Fram Centre, Tromsø, Norway
* evertmul@gmail.com

**Data Availability Statement:** All data files are available in the repository "opendata.uit.no" (dataverseNO), and are accessible through the following DOI: https://doi.org/10.18710/5WNWRL.

## Abstract

Tourism and nature-based recreation has changed dramatically during the COVID-19 pandemic. Travel restrictions caused sharp declines in visitation numbers, particularly in remote areas, such as northern Norway. In addition, the pandemic may have altered human-nature relationships by changing visitor behaviour and preferences. We studied visitor numbers and behaviour in northern Norway, based on user-generated data, in the form of photographic material that was uploaded to the popular online platform Flickr. A total of 195.200 photographs, taken by 5.247 photographers were subjected to Google's "Cloud Vision" automatic content analysis algorithm. The resulting collection of labels that were assigned to each photograph was analysed in structural topic models, using photography date (relative to the start of the pandemic measures in Norway) and reported or estimated photographers' nationality as explanatory variables. Our results show that nature-based recreation relating to "mountains" and "winter" became more prevalent during the pandemic, amongst both domestic and international photographers. Shifts in preferences due to the pandemic outbreak strongly depended on nationality, with domestic visitors demonstrating a wide interest in topics while international visitors maintained their preference for nature-based experiences. Among those activities that suffered the most from decline in international tourism was northern lights and cruises as indicated by the topic models. On the other hand, images depicting mountains and flora and fauna increased their prevalence during the pandemic. Domestic visitors, on the other hand, spent more time in urban settings as a result of restrictions, which results in a higher prevalence of non-nature related images. Our results underscore the need to consider the dynamic nature of human-nature relationships. The contrast in flexibility to adapt to changing conditions and travel restrictions should be incorporated in collaborative efforts of municipalities and tour operators to develop sustainable local nature-based tourism products, particularly in remote areas.

## Introduction

The global responses to the COVID-19 pandemic have had wide-reaching impacts on nature-based recreation and tourism [1, 2] and on human-nature interactions in general [3]. Since the

**Funding:** This study was financed by the Norwegian Research Council project BlueTrans- "Ocean Health under Blue Transitions", project no. 280778 The publication charges for this article have been funded by a grant from the publication fund of UiT The Arctic University of Norway. The funders had no role in study design, data collection and analysis, decision to publish, or preparation of the manuscript.

**Competing interests:** The authors have declared that no competing interests exist.

beginning of the pandemic in 2020, protected area visitations and wildlife tourism have decreased all over the world [4]. On average, the global number of international tourist arrivals in the world in 2020 was 73% lower compared to 2019. Between April and December 2020, the average reduction was more than 85%, compared to the same months in 2019 (World Tourism Organization United Nations, 2021) [5]. This decrease in visitations has a wide variety of consequences for both nature and society, ranging from job-loss [2], to fewer vehicle-wildlife collisions [6], behavioural changes in wildlife [7], increased opportunity for poaching [8] and a reduction in crowd-sourced scientific data, such as wildlife observations [9].

However, some nature areas experienced increasing numbers of visitors during the pandemic. Empirical data shows an increased number of visitations in areas that are easily accessible from urban areas, such as green spaces in cities [10, 11], or forests that are within a short distance from urban areas [12]. As a result of the pandemic, nature-recreationists opted to focus on nearby natural areas, as opposed to remote areas. For example, recreational bird-watchers increasingly visited nature areas closer to home [13]. This emphasizes the need to consider area-accessibility when evaluating effects of COVID-19 responses on nature-based tourism. Remote areas might experience different, or even antagonistic consequences, compared to areas that are easily accessible. To reduce the spread of COVID-19, many governments (including Norway) imposed local and international travel restrictions. As a result, remote areas throughout the world experienced drastic reductions in visitations [4].

Measuring the consequences of changes in visitation rate, or spatial and temporal recreational use in remote destinations can be challenging. Some studies have relied on user-surveys to evaluate trends in recreational activities [10, 13], whereas another avenue is to explore "big data" to empirically examine change in tourism and outdoor recreation patterns [14, 15]. In general, three forms of "Big data" can be identified: transaction data, device data, and user-generated data [16]. Transaction data refers to data that is generated from web searches, (online) bookings, or consumer cards. Device data includes data that is generated by mobile phones and other electronic devices, which for example provide information about the owners' location through for GPS, Bluetooth and WIFI [17]. Finally, user-generated data consists of information that is shared online by the users, such as through blogs, online reviews, or images that are posted on photo-sharing websites [16]. Rice & Pan [14] used an open-source Google dataset, based on aggregated location data from smartphones (device data), in order to determine changes in park visitations. Peng et al. [18] used web search data (transaction data) to predict the volume of tourism visitations in the Jiuzhai Valley (China). Other studies have employed data from social media platforms such as Twitter, Instagram and Flickr (user-generated data) to identify spatiotemporal trends in nature-based recreation at global and regional scales [19–21]. Flickr is particularly suitable for understanding spatial behaviour of nature-based tourism and recreation, as those using this platform are highly engaged in nature [22, 23]. Data from this platform has been shown to correspond to on-site visitor data on tourism and recreation. This type of data allows for automatic content analysis and provides opportunities for linking recreational use to e.g. country of origin [19, 20, 24].

In a user survey of 3.204 residents in Vermont in USA, Morse et al. [10] demonstrated that the spatial shifts and the non-material contribution of nature differs with respect to type of activities and socio-demographics. For instance, camping and social recreational activities, climbing and to some extent boating declined during the pandemic, whereas other activities such as gardening, hiking, jogging, and nature photography increased. Their results conform with previous studies that have found rural residents, domestic–and international visitors to differ in their spatial use of landscapes and the cultural ecosystem services they prefer [25, 26]. Differences between domestic and international visitors are particularly important for nature-based tourism destinations that rely upon domestic visitors to mitigate the decline of

international visitors. Spenceley et al. [4] suggest that tourism destinations need to draw on some of the lessons from the COVID-19 pandemic and build resilience by offering activities and products that appeal not only to international tourists, but also to domestic tourists.

In this study, we use automatic content analysis of pictures that were uploaded to the Flickr platform, combined with structured topic modelling, to examine the shift in spatial distribution of nature-based tourism and recreation in northern Norway, a remote nature-based tourism destination. This region experienced a boom in national and international visitation prior to the COVID-19 pandemic, in responds to recent active marketing efforts in the tourism sector [27]. However, in 2020, the international tourist arrivals in Norway decreased by 77%, compared to 2019, which is noticeably stronger than the European average decrease in visitations (-70%) or the global average (-73%) [5]. When considering only arrivals after March 2020 (when the Norwegian government first implemented travel restrictions), the decrease is more than 80%, compared to 2019. We investigate how appreciation of nature has shifted before and during the COVID-19 and explore the differences between domestic and international visitors. More specifically, we formulated four research questions:

1. How are visitation numbers influenced by the COVID-19 pandemic, based on photographers that were active on the Flickr platform?

2. What were the most popular photography topics in Northern Norway?

3. How did the pandemic affect the prevalence of these photographic topics amongst international and domestic photographers separately?

4. How did the prevalence of photography topics differ between international and domestic photographers, and was the compared prevalence influenced by the pandemic?

## Methods

### Data collection

The data that was used in this paper consists of a part of the dataset that was compiled by Runge et al. [20] and additional, more recent data. This dataset is based on photographs that users uploaded to the social media platform Flickr (www.flickr.com). Photos that were taken between 2001 and April 2021 in Northern Norway and which were uploaded to Flickr were processed using an automated content-identification algorithm: Google's Cloud Vision. Content Analysis can be defined as: "*an empirical (observational) and objective procedure for quantifying recorded "audio-visual" (including verbal) representation using reliable, explicitly defined categories ('values' on independent 'variables')*" [28]. Image Recognition Application Programming Interfaces (APIs), such as Google's Cloud Vision, are Automated Content Analysis tools that describe the content of photographic material in keywords, using machine learning algorithms to identify the objects in the photograph. Such APIs are particularly useful in the analysis of photographs from social media platforms (such as Flickr), as they convert pictures to a set of keywords that could be used to classify the content by use of textual analysis to indicate the rationale of taking the photo [24]. The initial dataset was extracted from Flickr on 4 December 2017 [20], while additional data was extracted between November 2020 and June 2021. Only photos that were geotagged (assigned to a location) were included in the dataset. Since this study is focused on northern Norway, we only extracted photos that were taken north of 65˚N, and between 0˚ and 35˚E. The study area covers the three northern most counties in Norway and the surrounding sea. The southern border therefore roughly corresponds to the southern border of Nordland county (Fig 1). Finally, we removed the remaining photos

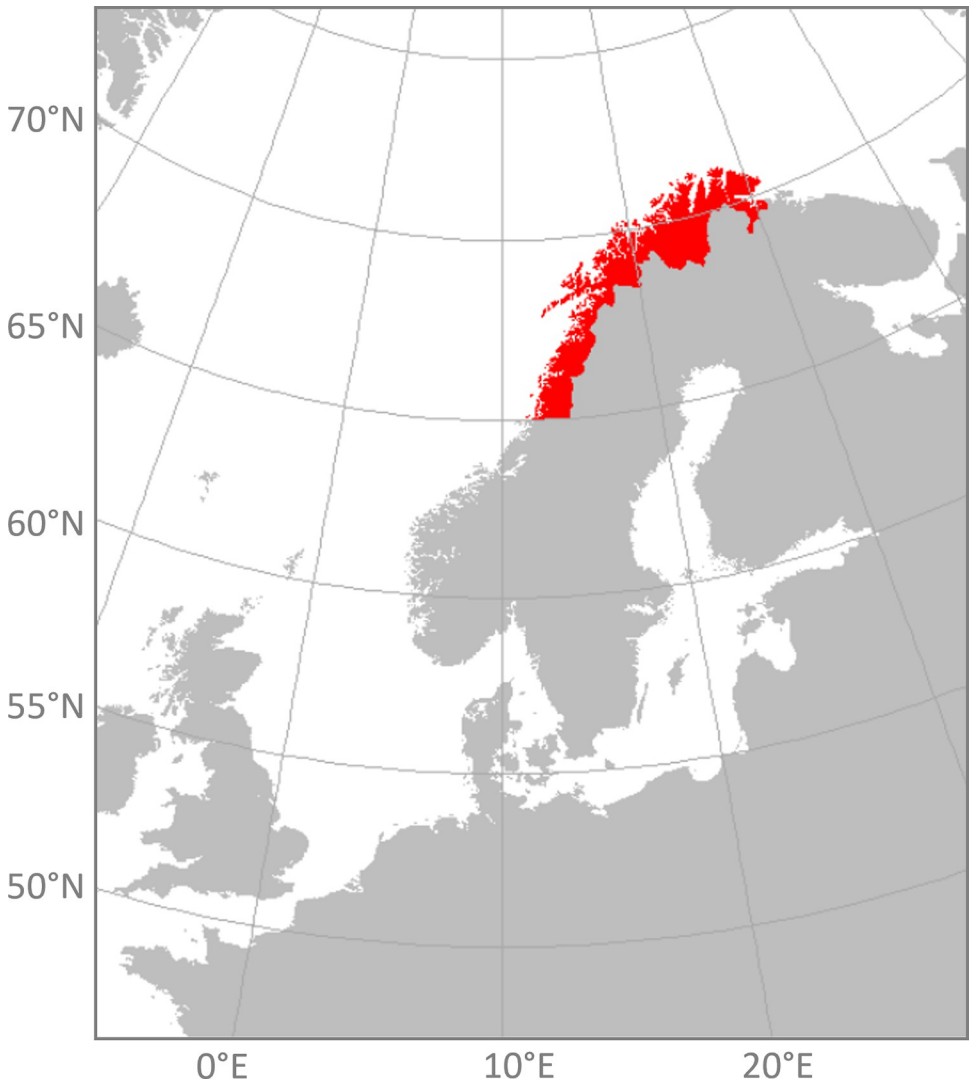

**Fig 1. Map of the study area (marked in red).**

that were taken in neighbouring countries (Finland & Sweden). We used "R" statistical computing software (version 4.0.3) for all data processing, filtering and further analysis [29]. We analysed the photos using the image recognition software package "Rooglevision" in R [30], which relies on Google's Cloud Vision algorithm to detect and classify the content of images. Using this approach, we described the content of each photo with a set of words (labels).

For each photograph, we stored only the photography date and location as well as a number representing each photographer. In addition, we stored up to 20 labels that were assigned to each photo by the Cloud Vision algorithm: we set the confidence threshold of the labelling algorithm to 50% to ensure an accurate description of the image contents. For each photographer, the country of origin was either extracted from the information the photographers shared on the Flickr platform, or, if no country of origin was shared, it was estimated based on the locations of the photographer's photos. For that purpose, we calculated the arithmetic mean of all geotagged photographs taken by the visitor, and evaluated if the photographer was either from Norway (domestic) or not from Norway (international).

As an initial exploration of the effects of the measures to prevent the spread of Covid-19, we studied the annual number of photographers, and the annual ratio between international and Norwegian photographers. Furthermore, we compared the percentage of Norwegian photographers before and after the implementation of the pandemic health measures.

## Structural topic model

The sets of keywords that describe each of the photographs can be regarded as bodies of texts (one for each photograph), which can be structured and classified. This process is known as Text (Data) Mining, which can be defined as: "*the application of algorithms and methods from the fields machine learning and statistics to texts with the goal of finding useful patterns.*" [31]. We used a Structural Topic Model (STM) to identify photography topics from the labels that have been assigned to each photograph [32]. An STM is a machine learning approach to identify unobserved groups of labels, or latent topics [33]. In principle, STMs estimate how associated each word is to each topic. At the same time, it estimates to which extent each picture is associated to each topic. Words can be associated to multiple topics, and pictures can be associated to multiple topics. The number of latent topics needs to be determined by the user beforehand. It is possible to compare statistical properties of models with a different number of topics to select the best model. However, since the purpose of these models is to evaluate the semantic coherence of a collection of texts, the strongest statistical models may not necessarily represent the best classification of latent topics [e.g. 34]. Instead, we opted to manually compare a range of models with a varying number of topics. Based on the most prevalent words within each topic, we arbitrary selected the model that contained the most coherent and distinguishable selection of topics, within the context of this study. Following this procedure, we selected a model with 8 topics. All spaces, punctuations, and special characters within each label were removed, to create a collection of singe-word "terms", which can be analysed by a STM. To improve the fit of the model, labels that occurred less than 10 times, or more than 90.000 times in the entire dataset were removed. The upper limit was selected to remove a single label: "sky" which was allocated to almost half of the photos by the Cloud Vision algorithm. As a result, this label has relatively little value in the identification of latent topics, as it fits in all topics. STMs were fitted using the "STM" package in R [35].

## Regression model

STM's differ from other topic models, as they can incorporate metadata to evaluate the influence of covariates in a regression-type analysis [35]. This enabled us to incorporate background information about the photographer (domestic or international) and information about when the photograph was taken (before or after the COVID-19 measures were implemented). As the covariate model structure influences the topic modelling procedure, there is no method to compare the statistical quality of models with different structures, as is common in other statistical models (e.g. by comparing the estimated prediction error, such as the AIC estimator). However, STMs can be used to study the effect of covariates on the distribution of words (labels) amongst the topics, based on a user-defined covariate model structure. Here we studied the interacting effect of two covariates: whether the picture was taken before or after implementation of COVID-19 measures, and whether the photographer was Norwegian (domestic) or international.

## Results

### Dataset and photographer country of origin

The dataset contains information from 195.200 photos that were taken in northern Norway by 5.247 photographers between 01-01-2000 and 31-03-2021 and uploaded to the Flickr website.

**Table 1. Annual number of international and Norwegian photographers.**

| | NUMBER OF PHOTOGRAPHERS | | | |
|---|---|---|---|---|
| YEAR | International | Norway | Total | %Change |
| 2000 | 8 | 4 | 12 | |
| 2001 | 7 | 3 | 10 | -17% |
| 2002 | 12 | 6 | 18 | 80% |
| 2003 | 16 | 6 | 22 | 22% |
| 2004 | 30 | 12 | 42 | 91% |
| 2005 | 36 | 16 | 52 | 24% |
| 2006 | 102 | 38 | 140 | 169% |
| 2007 | 156 | 71 | 227 | 62% |
| 2008 | 231 | 89 | 320 | 41% |
| 2009 | 276 | 109 | 385 | 20% |
| 2010 | 385 | 143 | 528 | 37% |
| 2011 | 550 | 170 | 720 | 36% |
| 2012 | 561 | 172 | 733 | 2% |
| 2013 | 644 | 157 | 801 | 9% |
| 2014 | 651 | 129 | 780 | -3% |
| 2015 | 617 | 136 | 753 | -3% |
| 2016 | 610 | 119 | 729 | -3% |
| 2017 | 543 | 104 | 647 | -11% |
| 2018 | 422 | 54 | 476 | -26% |
| 2019 | 383 | 46 | 429 | -10% |
| 2020 | 176 | 38 | 214 | -50% |

Note that the number of photos currently available on Flickr might differ, as users can remove or add pictures, or photos may be removed by the platform itself. Interannual variability in the number of photographs and photographers indicated an increase in the use of the Flickr platform that peaked in 2013, with 801 photographers (Table 1). The number of Norwegian photographers peaked in 2012 (172 photographers), while the number of international photographers peaked in 2014 (651 photographers). After 2014, the use of Flickr decreased steadily, as indicated by the decrease in total number of photographers. However, in 2020, this decline was amplified drastically, as the number of photographers decreased from 429 in 2019 to 205 in 2020 (50%), which is the lowest number of photographers since 2006.

The country of origin was known for 1.731 of the photographers (93.998 photos) and estimated for the remaining 3.516 photographers (101.202 photos). According to the downloaded and estimated photographer origins, the data consisted of 565 Norwegian and 4.682 non-Norwegian photographers. Overall, Norwegian photographers (11%) took approximately 16% of the photos (31.663). From 2019 to 2020, the number of international photographers dropped by 54%, while the number of Norwegian photographers dropped by only 17% in the same year. However, we have no explanation for the sharp reduction of 48% in the number of Norwegian photographers in the year 2018 (Table 1). We marked 1 April 2020 as the implementation date for the measures to prevent the spread of Covid-19 in Norway. Before that date, 11% of the photographers were Norwegian, while that percentage increased to 26% after the implementation.

## Google's cloud vision

Google's Cloud Vision algorithm assigned 6.215 unique labels to the photograph, 3.295 of which occurred less than 10 times, or more than 90.000 times. After removal of these labels,

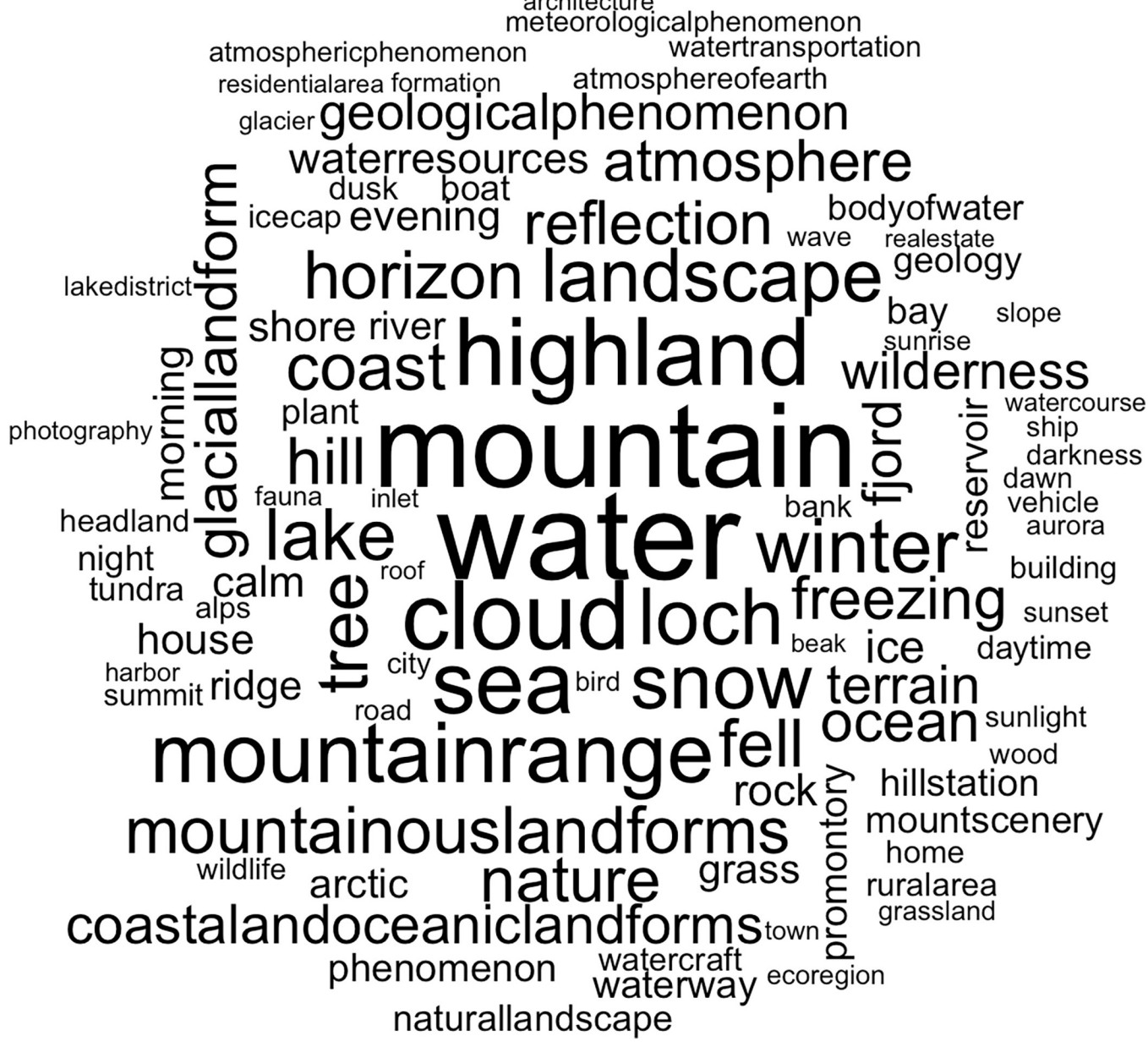

**Fig 2. Word cloud of all photograph labels.** The fond size represents the prevalence of each word in the dataset.

109 photographs had to be excluded from the dataset, since all the labels assigned to these photographs occurred less than 10 times throughout the dataset. The dataset that was analysed with the STM therefore contained information from 195.091 photographs. The most frequent label was "water", which was assigned to 63.600 photographs, followed by "mountain", "cloud", and "highland", which occurred respectively 59.909, 47.186, and 46.319 times (Fig 2).

## Structural topic model

An STM was used to identify 8 topics within the labels for each picture (Table 2). Despite the overlap between some of these topics, each topic can be distinguished from the other topics

**Table 2. Overall topic prevalence, and the 7 most prevalence words within each topic.**

| Topic | One-word description | Top 7 keywords | Expected proportions |
|---|---|---|---|
| 1 | Events | Event, Performance, Music, Entertainment, Performing Arts, Musician, Stage | 0.08 |
| 2 | Boats | Boat, Water, Watercraft, Waterway, Water transportation, Ship, Harbour | 0.08 |
| 3 | Water | Water, Sea, Loch, Lake, Horizon, Coast, Cloud | 0.22 |
| 4 | Night | Atmosphere, Night, Phenomenon, Darkness, Atmosphere of earth, Aurora, Nature | 0.13 |
| 5 | Mountains | Mountain, Highland, Landscape, Fell, Hill, Mountain range, Mountainous landforms | 0.14 |
| 6 | Winter | Snow, Winter, Freezing, Glacial landform, Geological phenomenon, Ice, Mountain range | 0.12 |
| 7 | Flora & Fauna | Grass, Plant, Tree, Wildlife, Bird, Grassland, Ecoregion | 0.11 |
| 8 | Urban | House, Home, Building, Road, Wood, Tree, Town | 0.12 |

based on sensible criteria. For example, topics 1, 2 and 8 all appear to be associated with human activities, in particular: arts & events (topic 1), ships & harbours (topic 2), and houses & towns (topic 8). Topics 3–7 are all related to nature, wildlife, and outdoor activities. These topics represent nature-based tourism in 5 different categories: water (topic 3), night & northern lights (topic 4), mountains (topic 5), snow & winter (topic 6), and plants & wildlife (topic 7). Based on the prevalence of keywords within each identified topic, we assigned a single-word description to each topic (Table 2). These descriptions are only a general representation of each topic, which may not appropriately capture each photo that was assigned to the topic. Some labels (e.g. water) occur in several topics. Overall, the most prevalent topic was topic 3 (water), while the prevalence of topics 4 to 8 was relatively equal (Table 2). Topics 1 (events) and 2 (boats) were the least prevalent, but each still contributed to approximately 8% of the data (Table 2).

## Influence of COVID-19 measures on photographer preference

We used the STM to study the influence of COVID-19 measures in Norway on the prevalence of photography topics. We marked April 1st, 2020 as the initiation of COVID-19 measures in Norway. After the implementation of COVID-19 measures, international photographers focused more on topics 5 (Mountains), 6 (Winter) and 7 (Flora & Fauna), while topics 1, 2, 3, and 4 (Events, Boats, Water, and Night) became less important (Fig 3). The change in prevalence for topic 7 (Flora & Fauna) was approximately 8%. Topic 8 (Urban) remained equally prevalent (no statistical difference). Amongst Norwegian photographers, topics 5, 6 and 8 (Mountains, Winter, and Urban) became more prevalent after COVID-19 measures were implemented, while topics 1 (Events) and 4 (Night) became less important after the implementation of COVID-19 measures. For Norwegian photographers, topics 2 (Boats), 3 (Water), and 7 (Flora & Fauna) remained equally prevalent (Fig 3).

Fig 3 illustrates the change in prevalence of each topic for international and domestic photographers separately, but it does not show the importance of topics for one group of photographers compared to the other group (international or domestic). Therefore, we also compared the influence of photographer's nationality on topic prevalence before and after COVID-19 (Fig 4). Topic 1 (Events) and topic 4 (Night) were more prevalent amongst Norwegian photographers, compared to international photographers, both before and after COVID-19 measures. Topics 3 (Water), 5 (Mountains), and 6 (Winter) were more prevalent amongst international photographers, both before and after COVID-19 measures. Topic 7 (Flora & Fauna) became more prevalent amongst international photographers after the COVID-19 measures. The opposite is true for topic 2 (Boats), which became more prevalent amongst Norwegian photographers after COVID-19 measures. After COVID-19 measures, topic 8 (Urban)

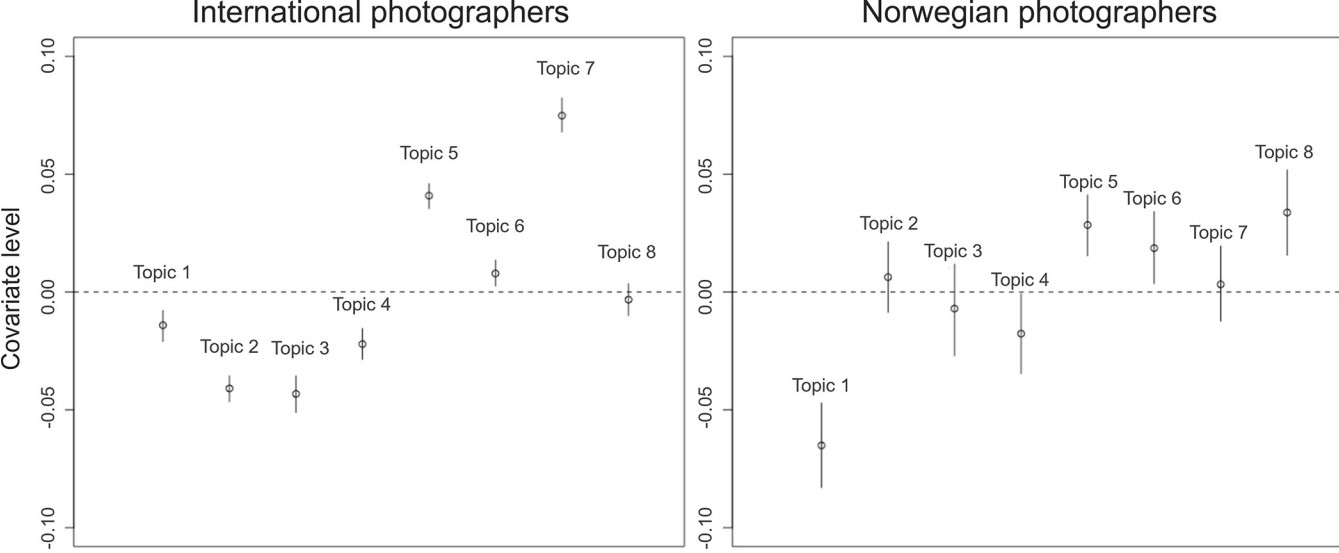

**Fig 3. Topic covariate levels (in proportions) after COVID-19 measures, compared to before COVID-19 measures.** Covariate levels > 0 indicate the topic was more prevalent after COVID-19 measures, compared to before the COVID-19 measures. The graph on the left illustrates changes in covariate levels after COVID-19 measures for international photographers, while the right graph illustrates changes in covariate levels for Norwegian photographers.

became more important for Norwegian photographers, compared to international photographers, while there had not been a significant difference before (Fig 4).

## Discussion

In this study, we use photographs from the social media platform Flickr, to determine how COVID-19 impacted the preferences and behaviour of visitors in northern Norway. The use of

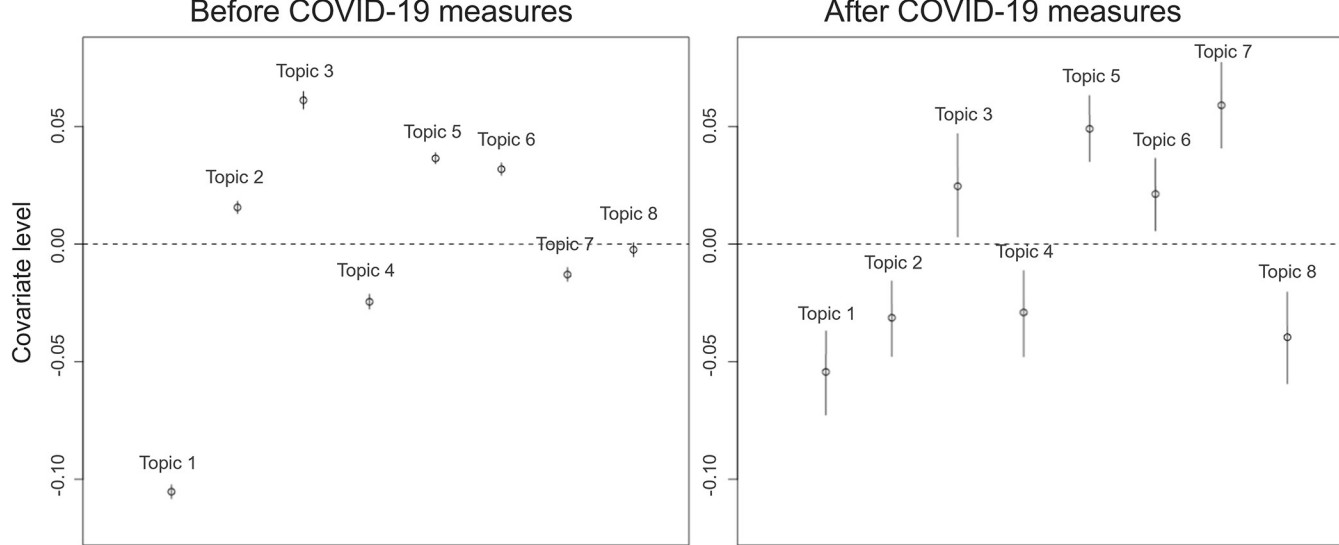

**Fig 4. Compared covariate levels for international photographers compared to Norwegians.** Covariate levels > 0 indicate that the topic was more popular (prevalent) amongst international photographers, while negative values indicate a prevalence amongst Norwegian photographers. The left graph shows estimated covariate levels before COVID-19 measures, while the graph on the right illustrates covariate levels after COVID-19 measures.

"big data" to understand the spatial heterogeneity in tourism preference and behaviour is currently still in its infancy [36, 37], and few authors have used such data to assess the impacts of COVID-19, depending on what visitors value when traveling to nature-based destinations such as northern Norway [15]. Here, we demonstrate that through a combination of automated content analysis and text data mining, "big data" can reveal continuous information on complex human-nature interactions, such as the impact of a global pandemic on visitor preferences in remote nature-based tourism destinations.

Norway is a popular destination for nature-based tourism [27]. This was clearly reflected in the preferences of photographers in northern Norway, as many of the most frequent keywords were associated with nature and outdoors activities. Overall, the "coastal" topic was the most prevalent topic, which is an indication of the importance of northern Norway's' extensive coastline for the tourism industry. Several studies have indicated that domestic and international visitors may hold different values [25, 26, 36]. For example, Muñoz et al (2019) [25] found that international visitors valued wilderness higher than domestic visitors to a natural park in southern Norway. The authors suggested these results could have been caused by differences in marketing strategy aimed towards visitors. These findings correspond to our results, which show that photography topics associated with nature or wilderness ("water", "mountains" and "winter"), were generally dominated by international photographers. In contrast, the topics "events", "nights" and "urban" were mostly associated with domestic photographers (Fig 4).

The onset of the global COVID-19 pandemic, and the measures to combat it's spread have severely impacted tourism and recreation around the world (Sigala, 2020) [2]. National and international measures hinder international and domestic travels, which leads to a reduction of the number of visitors, particularly in remote areas (Spenceley et al., 2021) [4]. Indeed, our results show a sharp decline in the annual number of photographers to post photographs of northern Norway, following the COVID-19 measures. Numbers of international visitors were affected more severely than domestic visitations, possibly caused by restricted mobility [38]. This is in line with official visitation numbers that indicate a particularly high reduction of international visitors in Northern Norway, compared to the European or global average [5].

The COVID-19 pandemic also affected the behaviour and preferences of visitors. During the pandemic, the overall prevalence for the nature-associated topics "mountains" and "winter" increased, while the topic "events" became less prevalent (Fig 3). Several studies have identified similar shifts in focus towards nature-based recreation [10, e.g. 12]. One potential explanation for such a change is that visitors experienced fewer opportunities to participate in collective activities during the pandemic [3]. In Norway, there are few restrictions on access to nature, and nature-based activities can be undertaken by individuals alone, without interaction with tour operators, nature guides, or other visitors. In contrast, the topic "events" inherently incorporates a social component (exhibitions, concerts, performances etc.), which has been restricted during the pandemic.

International and domestic photographers in northern Norway responded differently to the pandemic and COVID-19 measures. During the pandemic, international photographers took fewer photos within the topics "boats" and "water" (Fig 4), which could be explained by a decreased availability of organised or guided boat trips due to COVID-19 measures. At the same time, the accessibility to (private-owned) boats may not have been restricted as much for domestic photographers, which could explain why this topic remained equally prevalent during the pandemic. Similarly, domestic photographers showed an increased focus on the topic "urban" during the pandemic, while the prevalence did not change amongst international photographers. This topic does not exclusively refer to urban activities, but also includes the photography of buildings, houses, roads and even towns. Therefore, it is possible that domestic

photographers had access to alternative touristic activities during the pandemic, such as remote, private owned cabins to escape from the risk of infection in densely populated areas. International visitors did not appear to switch to these alternative touristic activities, as the availability of these type of activities was restricted for them.

While both international and domestic photographers appeared to appreciate nature-associated activities during the pandemic, the prevalence of the topic "Flora and Fauna" particularly increased amongst international photographers. There are a few possible explanations for the increase amongst international photographers. First, the topic "flora & fauna" can be enjoyed in relative isolation, as it does not necessarily depend on organised trips. Second, and in connection to this, the availability of recreational options that are associated to other photography topics may have been reduced during the pandemic. For example, options for organised boat trips may have been limited, due to regulations regarding social distance and maximum group size. Thirdly, the increase in prevalence of the topic "flora & fauna" amongst international photographers may be attributed to an increased interest in nature and wildlife, fuelled by the effects of the pandemic back home. Finally, the international photographers that visited northern Norway during the pandemic may not be representative of the international photographers that would visit northern Norway if there was no pandemic. As it is more difficult to enter Norway during the pandemic, this group of visitors is likely to be highly motivated to focus on the most meaningful attractions, which happen to be nature oriented for this core group of visitors that are willing to undergo all the travel restrictions, quarantine, and limited availability of organized tours, thus representing the core values of the visitor preferences in a remote area like northern Norway. Overall, these patterns suggest that under highly restrictive conditions, domestic visitors may be more flexible in finding alternative touristic activities, while international visitors may be relatively rigid in the consumption of nature-based products, especially in remote areas where the availability of alternative (e.g. cultural) attractions are sparse.

## Study limitations

"Big Data", generated by social media provides valuable information on human-nature interactions, within the context of recreation and tourism [39]. Nonetheless, it comes with a set of limitations that might influence study results. For example, there may be a bias in the study sample, as not everyone uses social media in the same way and to the same extent. In our study, Norwegian domestic tourists appear to be underrepresented on the Flickr platform. Based on the number of photographers in this study, domestic visitors formed less than a quarter of the total number of visitors in northern Norway, during the last decade. Some of the domestic photographers may have been identified erroneously as international visitors, as the nationality of some photographers had to be estimated from the geographic distribution of their photographs. It is possible that a well-travelled Norwegian photographer is mis-identified as an international photographer. One explanation may be that the perceived value of a travel is influenced by the travel distance [39]. Specifically, visitors that stay relatively close to home may value a travel destination less, compared to visitors that travelled a long distance. This means that Norwegian photographers might be less inclined to take and share photographs of their travels within Norway. However, big data from social media was found to accurately represent values and preferences of visitors in several studies [23]. It is therefore unlikely that the relatively low number of Norwegian photographers causes a misrepresentation of the values and preferences of domestic travellers in Norway in general.

Our results indicate that under highly restricted conditions, both international and domestic visitors shift towards tourism activities that may not be dependent on tour operators. This

means an additional pressure for the tourism industry in a remote area such as northern Norway, besides the pressure of reduced visitor numbers. Alternatively, our results may be caused by a reduction in the offer of organised tourism activities during the pandemic, which would stimulate visitors to find activities that can be done independently. In either case, this study warrants a careful evaluation of the role of the nature-based tourism industry in remote areas, to ensure the sustainability of the industry in the future.

In conclusion, our study indicates that human-nature relationships could be highly dynamic depending on a global pandemic or other crises that reduces travel to remote nature-based destinations. Whereas big data analytics have previously documented pandemic impacts on tourism numbers on a global scale [e.g. 15], our study demonstrates how pictures taken by visitors emphasized the importance of individual access to nature versus collective and organized activities during a pandemic. This was particularly important for international tourists, whereas domestic visitors have a high capacity to adapt to restrictive environments under the pandemic scenario and shift from nature-based activities to urban activities. This highlights the urgency of collaborative efforts between local municipalities and tour operators to ensure the long-term resilience of tourism destinations.

## Author Contributions

**Conceptualization:** Evert Mul, Francisco Javier Ancin Murguzur, Vera Helene Hausner.

**Formal analysis:** Evert Mul, Francisco Javier Ancin Murguzur.

**Funding acquisition:** Vera Helene Hausner.

**Investigation:** Francisco Javier Ancin Murguzur, Vera Helene Hausner.

**Methodology:** Evert Mul, Francisco Javier Ancin Murguzur, Vera Helene Hausner.

**Project administration:** Francisco Javier Ancin Murguzur.

**Resources:** Francisco Javier Ancin Murguzur, Vera Helene Hausner.

**Supervision:** Vera Helene Hausner.

**Validation:** Evert Mul, Francisco Javier Ancin Murguzur, Vera Helene Hausner.

**Visualization:** Evert Mul.

**Writing – original draft:** Evert Mul.

**Writing – review & editing:** Evert Mul, Francisco Javier Ancin Murguzur, Vera Helene Hausner.

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
