## [Decision Letter · Decision Letter 0]

3 May 2022

PONE-D-22-04631Impact of the COVID-19 pandemic on human-nature relations in a remote nature-based tourism destinationPLOS ONE

Dear Dr. Mul,

Thank you for submitting your manuscript to PLOS ONE. After careful consideration, we feel that it has merit but does not fully meet PLOS ONE’s publication criteria as it currently stands. Therefore, we invite you to submit a revised version of the manuscript that addresses the points raised during the review process.

We look forward to receiving your revised manuscript.

Kind regards,

Daniel Capella Zanotta

Academic Editor

PLOS ONE

Journal Requirements:

Reviewers' comments:

Reviewer's Responses to Questions

**Comments to the Author**

1. Is the manuscript technically sound, and do the data support the conclusions?

Reviewer #1: Yes

Reviewer #2: Yes

Reviewer #3: Yes

Reviewer #4: Yes

2. Has the statistical analysis been performed appropriately and rigorously? 

Reviewer #1: Yes

Reviewer #2: Yes

Reviewer #3: I Don't Know

Reviewer #4: Yes

3. Have the authors made all data underlying the findings in their manuscript fully available?

Reviewer #1: Yes

Reviewer #2: Yes

Reviewer #3: Yes

Reviewer #4: Yes

4. Is the manuscript presented in an intelligible fashion and written in standard English?

Reviewer #1: Yes

Reviewer #2: Yes

Reviewer #3: Yes

Reviewer #4: Yes

5. Review Comments to the Author

Reviewer #1: I accept without any hesitation. The topic is especially important. Well-structured paper. Methodology is correct. The literature added is complete. Results are useful for scientist and researchers as well. I recommend publishing this paper. Native English proofreading is suggested.

Reviewer #2: Thank you for the opportunity to review the article " Impact of the COVID-19 pandemic on human-nature relations in a remote nature-based tourism destination".

Examining tourism and nature-based recreation is an important area of research, particularly bearing in mind the current relevance of COVID-19. At first sight, the quality of ideas and methods of this paper is fairly adequate. Moreover, in general, the article is well structured and easy to read. However, there are several aspects that I ponder could add to improve the quality of the manuscript. From a general perspective, my main concerns are the real impact of COVID-19 pandemic on human-natured relations, that seems a bit vague in my opinion. Besides of this, there are several issues that could also be amended in order to enhance the quality and relevance of the paper.

Regarding Table 1and Table 2, each table should be preferably be on one page.

The contributions of the paper is unclear. What are contributions to previous literatures (Theoretical implications)? Therefore, the limitations of this study should be pointed out.

Reviewer #3: The article is very interesting. Both the sampling (time and number of photographs/photographers) and the methods (statistics, database, cross-references and content analysis) contributed to an unprecedented approach, as far as I am aware, of a recurring subject: photographs in social media. In addition, the comparison with the COVID-19 pandemic allowed a more recent view of the impact of the pandemic on travel and consequently on the environment (positively) and socio-economy (negatively) of tourist destinations. As an improvement I recommend:

- improve the theoretical framework a little, inserting direct quotes of some concepts, such as "destination marketing", "tourist imagery" and the like;

- improve the method a little, bringing an author who defines the "content analysis" and bring a map of location and access of the study area;

- improve the discussion of the results by bringing in more authors, both theoretical (concepts addressed in the categorizations of the photographs) and practical (case studies that analyze Flickr, Instagram, Tripadvisor, Facebook and other social networks; regions with a more arctic/antarctic climate; attractions similar to registered). Congratulations to the authors for the research. Unfortunately, I don't have any considerations about the statistical use, because I don't have the know-how or expertise.

Reviewer #4: The authors of the manuscript: "Impact of the COVID-19 pandemic on human-nature relations in a remote nature-based tourism destination" reported a well-performed study that focuses to evaluate the influence of the last pandemic on tourism, mainly on how an appreciation of nature has shifted before and during the COVID-19 and explore the differences between domestic and international visitors in Northern Norway. The authors used the Flickr platform to obtain the necessary photographs and then applied a Structural Topic Model, with labels assigned to each photograph, and then performed a regression-type analysis.

In general, the data obtained and reported in the manuscript are well corroborated and discussed. The manuscript is concise and the appropriate references are cited.

The authors need to address the below comments to strengthen the quality of the manuscript:

-To correct the grammar and typos mistakes.

-To include data with values in the Abstract.

-To use the abbreviation for Structural Topic Model, STM, first when appearing in the text.

-To highlight the limits of this study.

-Increase the label fonts in Figs 2 and 3.

6. PLOS authors have the option to publish the peer review history of their article (what does this mean?). If published, this will include your full peer review and any attached files.

Reviewer #1: No

Reviewer #2: No

Reviewer #3: **Yes: **Ricardo Eustáquio Fonseca Filho

Reviewer #4: No

---

## [Author Response · Author response to Decision Letter 0]

24 Jun 2022

Response to reviewers

Reviewer #1: “I accept without any hesitation. The topic is especially important. Well-structured paper. Methodology is correct. The literature added is complete. Results are useful for scientist and researchers as well. I recommend publishing this paper. Native English proofreading is suggested.”

We thank reviewer #1 for his or her recommendation to publish the manuscript without hesitation. In accordance to the reviewers suggestion for proofreading, we have made sure to thoroughly proof-read the manuscript before resubmitting, and we have addressed grammar and spelling errors in the text. 

Reviewer #2: “Thank you for the opportunity to review the article " Impact of the COVID-19 pandemic on human-nature relations in a remote nature-based tourism destination".

Examining tourism and nature-based recreation is an important area of research, particularly bearing in mind the current relevance of COVID-19. At first sight, the quality of ideas and methods of this paper is fairly adequate. Moreover, in general, the article is well structured and easy to read. However, there are several aspects that I ponder could add to improve the quality of the manuscript. From a general perspective, my main concerns are the real impact of COVID-19 pandemic on human-natured relations, that seems a bit vague in my opinion. Besides of this, there are several issues that could also be amended in order to enhance the quality and relevance of the paper.

Regarding Table 1and Table 2, each table should be preferably be on one page.

The contributions of the paper is unclear. What are contributions to previous literatures (Theoretical implications)? Therefore, the limitations of this study should be pointed out.”

We thank reviewer #2 for the thorough evaluation of our manuscript, and for the valuable suggestions. In accordance to these suggestions, we have made the following general revisions:

1) We revised our discussion of the impact of COVID-19 on human-nature relations, by highlighting the differences in dependency on individual access between international and domestic visitors. Furthermore, we have expanded the theoretical context by bringing in more authors on human-nature relations in the discussion section.

2) Tables 1 and 2 will be adjusted in collaboration with the editor so that they each fit on one page

3) To discuss the limitations of this study in more detail, we added a paragraph on study limitations to the discussion. 

Furthermore, we have made the following revisions, based on the reviewers’ in-text suggestions:

Reviewer #2 in-text comments:

L23-26: We argue that this difference between domestic and international visitors is a reflection of the difference in flexibility and independence between these two groups. We have discussed this reasoning in more detail in the discussion section. 

L45: following the suggestion of the reviewer, we have removed the words “calculated from”.

L60: We agree with the reviewer that it may be important to specifically mention that Norway is among the countries that imposed travel restrictions, we therefore added “(including Norway)” to the sentence. 

L68: We thank the reviewer for this compliment.

L90-91: The reviewer rightfully notes that the set of activities that declined during the pandemic mostly consists of collective activities, while the increased activities include mostly activities that can be done individually. We opted not to further emphasize this contrast here in the introduction, as it is further discussed in the discussion section. 

L92: Based on the comment of the reviewer, we decided to omit this sentence altogether. 

L96: The reviewer notes that no clear differences between domestic and international visitor preferences have been described in the literature that is cited here. However, our aim with this sentence is to highlight the importance of potential differences. We therefore slightly adjusted the sentence, which now reads: 

“Differences between domestic and international visitors are particularly important for nature-based tourism destinations …” 

L100: The reviewer is correct in the statement that the mentioned lessons that could be learned from the pandemic focus on local to global recovery, and that it will be difficult to offer activities and products for both at the same time. Here we would like to emphasize that domestic visitors (local recovery) are sometimes overlooked, and that the focus is entirely on international visitors. We have changed the sentence to clarify this point more precisely: 

“… offering activities and products that appeal not only to international tourists, but also to domestic tourists.” 

L104-105: Following the suggestion from the reviewer, we have added a reason behind the pre-pandemic tourism boom: 

“…, in responds to recent active marketing efforts in the tourism sector. 

L108: We thank the reviewer for this clarification, and we have changed the acronym accordingly. 

L109: The reviewer is correct, and we have changed the sentence to: 

“(when the Norwegian government first implemented travel restrictions)”

L112: We thank the reviewer for this compliment.

L121: Following the suggestion from the reviewer, we have added a map of the region (Figure 1).

L122: We thank the reviewer for this compliment.

L126: We thank the reviewer for this compliment. 

L130: Since the dataset used in this study is an expansion of the dataset that was compiled by Runge et al, 2020, we have opted not to include these statistics in this paper.

L132: We have added the following sentence: 

“The study area covers the three northern most counties in Norway and the surrounding sea. The southern border therefore roughly corresponds to the southern border of Nordland county (Figure 1).”

L136: Following the reviewers’ suggestion, we expanded our explanation of the software used as follows: 

“We analysed the photos using the image recognition software package “Rooglevision” in R (Teschner, 2021), which relies on Google’s Cloud Vision algorithm to detect and classify the content of images. With this approach, we described the content of each photo with a set of words (labels).” 

L138: No filter was used to reduce the large number of labels.

L141: We agree with the reviewer that this is interesting.

L143: The reviewer rightfully questions our choice of using the first-person plural form to describe our approach, rather than the third-person singular form. We opted to use this form, as the use of the third-person singular may lead to confusion in some occasions in this manuscript, as we also use the third-person singular form to describe the decision making process of the photographers. We therefore prefer to continue using our current format, unless it is a style requirement of the journal to use the third-person singular form. 

L148: We adopted the reviewers suggestion; the sentence now reads:

“… before and after the implementation of the pandemic health measures.”

L149: We thank the reviewer for this compliment. 

L150 & L151: We agree with the reviewer, and we now use the acronym STM (after the first use), rather than the full word Structural Topic Model. 

L167 & 168: This value was selected to remove the word “sky”, which was in almost all of the picture, and therefore not considered to be a relevant description.

L182-183: We thank the reviewer for this compliment.

L183: This is a good point raised by the reviewer, and we have adapted a more concise terminology, not only here, but throughout the manuscript. We now refer to the disease behind this pandemic as COVID-19. 

L185: We decided not to include original photographs to the manuscript, since the focus of this study was not to evaluate the content analysis algorithm. The original photographic material was not relevant for the main analysis (Structural Topic Modelling) which only used the labels that were assigned to the photographs by the algorithm. 

L187 & 188: We thank the reviewer for this compliment.

L189: We have changed the sentence in agreement with the reviewer:

“Note that the number of photos currently available on Flickr might differ, as users can remove or add pictures, or photos may be removed by the platform itself.”

L192: We were not able to find a substantiated reason behind the decrease in the number of photographs on Flickr after 2014. Any reason we could find (popularity of other platforms perhaps) is mostly based on speculation.

L192-193: We agree with the reviewer that this sentence is somewhat redundant, as it is implied in the previous sentence. However, we opted to leave it in, as it connects to the sentence that follows.

L194: We agree with the reviewer that it can be interesting to add to the text the number of photographers in the “best” year, in comparison with the number of photographers in the “worst” year. We have added this to one of the previous sentences, which now reads: 

“Interannual variability in the number of photographs and photographers indicated an increase in the use of the Flickr platform that peaked in 2013, with 801 photographers (Table 1). The number of Norwegian photographers peaked in 2012 (172 photographers), while the number of international photographers peaked in 2014 (651 photographers).”

L195-196: We follow the reviewers’ suggestion to remove the data from 2021 from the table, as this was an incomplete year. 

L197: Following the reviewers’ suggestion, we have added the year with the most Norwegian and the most international photographers in the text (see the comment for L194).

L204-207: We agree that it is interesting that the sanitary protocols of the pandemic affect international photographers more than the local/national photographers. Following the reviewers’ suggestion, we have added the percentage Norwegian photographers to the sentence:

“Overall, Norwegian photographers (11%) took approximately 16 % of the photos (31.663).”

L218-219: It is true that there appears to be an emphasis on ecotourism. We addressed this point in the discussion, by describing the importance and popularity of nature-based tourism in Norway. 

L221: We thank the reviewer for this compliment, and we have corrected the spelling error.

L223: We thank the reviewer for this compliment.

L237/table2 (1): The reviewer asks if the software used to run the STM in this study relies on the Content Analysis method. This is not the case, as STMs are a form of Text Mining, rather than Content Analysis. However, Content Analysis was used by the image recognition software to describe the content of the photographs.

The reviewer raises a very important point: we have not yet sufficiently introduced theoretical concepts, such as Text Mining and (Automated) Content Analysis. We therefore expanded the theoretical framework in the methods section, by bringing in authors to define these concepts. The following two paragraphs were added:

“Content Analysis can be defined as: “an empirical (observational) and objective procedure for quantifying recorded “audio-visual” (including verbal) representation using reliable, explicitly defined categories (‘values’ on independent ‘variables’)” (Bell, 2011). Image Recognition Application Programming Interfaces (APIs), such as Google’s Cloud Vision, are Automated Content Analysis tools that describe the content of photographic material in keywords, using machine learning algorithms to identify the objects in the photograph. Such APIs are particularly useful in the analysis of photographs from social media platforms (such as Flickr), as they convert pictures to a set of keywords that could be used to classify the content by use of textual analysis to indicate the rationale of taking the photo (Richards & Tunçer, 2018).” 

“The sets of keywords that describe each of the photographs can be regarded as bodies of texts (one for each photograph), which can be structured and classified. This process is known as Text (Data) Mining, which can be defined as: “the application of algorithms and methods from the fields machine learning and statistics to texts with the goal of finding useful patterns.” (Hotho et al., 2005).”

L237/table2 (2): Topic 8 (Urban) indeed contains the word “tree” as one of its key words, however, the other keywords in this topic all refer to houses, buildings, towns etc. which is why we refer to this topic as “Urban”. Words (or labels) such as “tree” occur in multiple topics, but the topics are not distinguished, based on unique words, but on the collective occurrence of the key words. If a photo is described by the words “tree”, “birds”, and “wildlife”, it can be assigned to a different topic than when it is described by the words “tree”, “road”, “house”. 

L239 & 241 & 247: We agree with the reviewer, and we have replaced the word “corona” with “COVID-19” throughout the manuscript. 

L272: We thank the reviewer for the suggestion to include more theory and quotes in the discussion section, and we have re-written the discussion, in order to bring more authors in the discussion. 

L290: Following the suggestion from the reviewer, we have re-read and adjusted the discussion section to avoid repetition. 

Reviewer #3: The article is very interesting. Both the sampling (time and number of photographs/photographers) and the methods (statistics, database, cross-references and content analysis) contributed to an unprecedented approach, as far as I am aware, of a recurring subject: photographs in social media. In addition, the comparison with the COVID-19 pandemic allowed a more recent view of the impact of the pandemic on travel and consequently on the environment (positively) and socio-economy (negatively) of tourist destinations. As an improvement I recommend:

- improve the theoretical framework a little, inserting direct quotes of some concepts, such as "destination marketing", "tourist imagery" and the like;

- improve the method a little, bringing an author who defines the "content analysis" and bring a map of location and access of the study area;

- improve the discussion of the results by bringing in more authors, both theoretical (concepts addressed in the categorizations of the photographs) and practical (case studies that analyze Flickr, Instagram, Tripadvisor, Facebook and other social networks; regions with a more arctic/antarctic climate; attractions similar to registered). Congratulations to the authors for the research. Unfortunately, I don't have any considerations about the statistical use, because I don't have the know-how or expertise.

We thank reviewer #3 for his suggestions for improvement of the manuscript. 

Following the suggestions of the reviewer, we have expanded the method section, by adding a map of the area, and by bringing in authors to define the concept of (Automated) Content Analysis, as well as the concept of Text Mining. The following two paragraphs were added: 

“Content Analysis can be defined as: “an empirical (observational) and objective procedure for quantifying recorded “audio-visual” (including verbal) representation using reliable, explicitly defined categories (‘values’ on independent ‘variables’)” (Bell, 2011). Image Recognition Application Programming Interfaces (APIs), such as Google’s Cloud Vision, are Automated Content Analysis tools that describe the content of photographic material in keywords, using machine learning algorithms to identify the objects in the photograph. Such APIs are particularly useful in the analysis of photographs from social media platforms (such as Flickr), as they convert pictures to a set of keywords that could be used to classify the content by use of textual analysis to indicate the rationale of taking the photo (Richards & Tunçer, 2018).”

“The sets of keywords that describe each of the photographs can be regarded as bodies of texts (one for each photograph), which can be structured and classified. This process is known as Text (Data) Mining, which can be defined as: “the application of algorithms and methods from the fields machine learning and statistics to texts with the goal of finding useful patterns.” (Hotho et al., 2005).”

Furthermore, we have revised the discussion section, based on the suggestions of the reviewer. Specifically, we have introduced more authors and we have linked our results to other practical studies. 

Although we have expanded the theoretical framework of human-nature relationships, specifically in the discussion section, we have opted not to expand on the concepts "destination marketing" and "tourist imagery". Destination marketing and tourism imagery are not completely within the scope of this paper as the focus in on the dynamics of human-nature relations under a pandemic outbreak, using big data analytics. The novelty of this paper is reflected by the automatic content analysis combined with structural topic modelling to discern complex responses relating to different kinds of nature activities enjoyed, individual access versus organized tours, and domestic and international preferences. We have added references to other authors who have mapped these complexities in human-nature relationships using big data analytics, but we cannot switch the focus towards completely alternative theories, as the rationale of using big data analytics is to discover emerging patterns rather than to test theory.

Reviewer #4: The authors of the manuscript: "Impact of the COVID-19 pandemic on human-nature relations in a remote nature-based tourism destination" reported a well-performed study that focuses to evaluate the influence of the last pandemic on tourism, mainly on how an appreciation of nature has shifted before and during the COVID-19 and explore the differences between domestic and international visitors in Northern Norway. The authors used the Flickr platform to obtain the necessary photographs and then applied a Structural Topic Model, with labels assigned to each photograph, and then performed a regression-type analysis.

In general, the data obtained and reported in the manuscript are well corroborated and discussed. The manuscript is concise and the appropriate references are cited.

The authors need to address the below comments to strengthen the quality of the manuscript:

-To correct the grammar and typos mistakes.

-To include data with values in the Abstract.

-To use the abbreviation for Structural Topic Model, STM, first when appearing in the text.

-To highlight the limits of this study.

-Increase the label fonts in Figs 2 and 3.

We thank reviewer #4 for his or her suggestions to strengthen the quality of the manuscript. In accordance to these suggestions, we have carefully evaluated and corrected the grammar and spelling errors in the manuscript, we have adopted the reviewers’ suggestion to use the abbreviation for Structural Topic Models after the firs appearance in the text, and we have increased the label fonts in Figures 2 and 3 (now figures 3 and 4). Furthermore, we have added data with values regarding the number of photographs and photographers to the abstract.

Finally we added the following paragraph to the discussion section, regarding the limits of this study, following the reviewers’ suggestion:

“Big Data”, generated by social media provides valuable information on human-nature interactions, within the context of recreation and tourism (Wood et al., 2013). Nonetheless, it comes with a set of limitations that might influence study results. For example, there may be a bias in the study sample, as not everyone uses social media in the same way and to the same extent. In our study, Norwegian domestic tourists appear to be underrepresented on the Flickr platform. Based on the number of photographers in this study, domestic visitors formed less than a quarter of the total number of visitors in northern Norway, during the last decade. Some of the domestic photographers may have been identified erroneously as international visitors, as the nationality of some photographers had to be estimated from the geographic distribution of their photographs. It is possible that a well-travelled Norwegian photographer is mis-identified as an international photographer. One explanation may be that the perceived value of a travel is influenced by the travel distance (Wood et al., 2013). Specifically, visitors that stay relatively close to home may value a travel destination less, compared to visitors that travelled a long distance. This means that Norwegian photographers might be less inclined to take and share photographs of their travels within Norway. However, big data from social media was found to accurately represent values and preferences of visitors in several studies (Toivonen et al., 2019). It is therefore unlikely that the relatively low number of Norwegian photographers causes a misrepresentation of the values and preferences of domestic travellers in Norway in general.”

---

## [Decision Letter · Decision Letter 1]

8 Aug 2022

Impact of the COVID-19 pandemic on human-nature relations in a remote nature-based tourism destination

PONE-D-22-04631R1

Dear Dr. Mul,

We’re pleased to inform you that your manuscript has been judged scientifically suitable for publication and will be formally accepted for publication once it meets all outstanding technical requirements.

Kind regards,

Daniel Capella Zanotta

Academic Editor

PLOS ONE

Additional Editor Comments (optional):

Reviewers' comments:

Reviewer's Responses to Questions

**Comments to the Author**

1. If the authors have adequately addressed your comments raised in a previous round of review and you feel that this manuscript is now acceptable for publication, you may indicate that here to bypass the “Comments to the Author” section, enter your conflict of interest statement in the “Confidential to Editor” section, and submit your "Accept" recommendation.

Reviewer #4: All comments have been addressed

2. Is the manuscript technically sound, and do the data support the conclusions?

Reviewer #4: Yes

3. Has the statistical analysis been performed appropriately and rigorously? 

Reviewer #4: Yes

4. Have the authors made all data underlying the findings in their manuscript fully available?

Reviewer #4: Yes

5. Is the manuscript presented in an intelligible fashion and written in standard English?

Reviewer #4: Yes

6. Review Comments to the Author

Reviewer #4: In my opinion, the present version of the manuscript can be accepted for publication in Plos One, as the authors made all the requirements to improve the MS.

7. PLOS authors have the option to publish the peer review history of their article (what does this mean?). If published, this will include your full peer review and any attached files.

Reviewer #4: No

---

## [Editor Report · Acceptance letter]

20 Sep 2022

PONE-D-22-04631R1 

Impact of the COVID-19 pandemic on human-nature relations in a remote nature-based tourism destination 

Dear Dr. Mul:

I'm pleased to inform you that your manuscript has been deemed suitable for publication in PLOS ONE. Congratulations! Your manuscript is now with our production department. 

Kind regards, 

on behalf of

Dr. Daniel Capella Zanotta 

Academic Editor

PLOS ONE